# Complex patterns of multimorbidity associated with severe COVID-19 and long COVID
Maik Pietzner [1,2,3] ✉, Spiros Denaxas[4,5,6,7], Summaira Yasmeen[1], Maria A. Ulmer [8],
Tomoko Nakanishi [2], Matthias Arnold [8,9], Gabi Kastenmüller [8], Harry Hemingway [4,5,7,10] ✉ &
Claudia Langenberg [1,2,3,10] ✉

## Abstract

**Background** Early evidence that patients with (multiple) pre-existing diseases are at highest risk for severe COVID-19 has been instrumental in the pandemic to allocate critical care resources and later vaccination schemes. However, systematic studies exploring the breadth of medical diagnoses are scarce but may help to understand severe COVID-19 among patients at supposedly low risk.

**Methods** We systematically harmonized >12 million primary care and hospitalisation health records from ~500,000 UK Biobank participants into 1448 collated disease terms to systematically identify diseases predisposing to severe COVID-19 (requiring hospitalisation or death) and its post-acute sequalae, Long COVID.

**Results** Here we identify 679 diseases associated with an increased risk for severe COVID-19 (n = 672) and/or Long COVID (n = 72) that span almost all clinical specialties and are strongly enriched in clusters of cardio-respiratory and endocrine-renal diseases. For 57 diseases, we establish consistent evidence to predispose to severe COVID-19 based on survival and genetic susceptibility analyses. This includes a possible role of symptoms of malaise and fatigue as a so far largely overlooked risk factor for severe COVID-19. We finally observe partially opposing risk estimates at known risk loci for severe COVID-19 for etiologically related diseases, such as post-inflammatory pulmonary fibrosis or rheumatoid arthritis, possibly indicating a segregation of disease mechanisms.

**Conclusions** Our results provide a unique reference that demonstrates how 1) complex co-occurrence of multiple – including non-fatal – conditions predispose to increased COVID-19 severity and 2) how incorporating the whole breadth of medical diagnosis can guide the interpretation of genetic risk loci.

## Plain Language Summary

Early in the COVID-19 pandemic it was clear that people with multiple chronic diseases were vulnerable and needed special protection, such as shielding. However, many people without such diseases required hospital care or died from COVID-19. Here, we investigated the importance of underlying diseases, including mild diseases not requiring hospitalization, for COVID-19 outcomes. Using information from electronic health records we find that many severe, but also less severe diseases increase the risk for severe COVID-19 and its impact on health even months after acute infection (Long COVID). This included an almost two-fold higher risk among people that reported poor well-being and fatigue. Our findings show the value of using primary care health records and the need to consider all the medical history of patients to identify those in need of special protection.

From the outset of the COVID-19 pandemic it was evident that underlying conditions were associated with both the risk of infection with SARS-CoV-2, the cause of COVID-19, and the risk of it being severe, based on the risk of hospitalisation, to ventilation and death[1]. Initial focus was on the small number of diseases known to put people at higher risk of other respiratory viral infections, such as influenza. The Center for Disease Control in the US and other national bodies published lists of diseases associated with COVID-19 and in the UK more than 1 million people were identified as

[1]Computational Medicine, Berlin Institute of Health at Charité – Universitätsmedizin Berlin, Berlin, Germany. [2]Precision Healthcare University Research Institute, Queen Mary University of London, London, UK. [3]MRC Epidemiology Unit, University of Cambridge, Cambridge, UK. [4]Institute of Health Informatics, University College London, London, UK. [5]Health Data Research UK, London, UK. [6]British Heart Foundation Data Science Centre, London, UK. [7]National Institute of Health Research University College London Hospitals Biomedical Research Centre, London, UK. [8]Institute of Computational Biology, Helmholtz Zentrum München - German Research Center for Environmental Health, Neuherberg, Germany. [9]Department of Psychiatry and Behavioral Sciences, Duke University, Durham, NC, USA. [10]These authors contributed equally: Harry Hemingway, Claudia Langenberg. ✉e-mail: maik.pietzner@bih-charite.de; h.hemingway@ucl.ac.uk; claudia.langenberg@qmul.ac.uk

clinically extremely vulnerable and required 'shielding' based on having one or more specified diseases[2]. This included older individuals, men, and those with the presence of multiple, pre-exiting long-term conditions, such as impaired immunity, type 2 diabetes, hypertension, or chronic kidney disease (CKD)[1].

However, the vast body of COVID-19 risk factor studies were based on a candidate approach (e.g., diseases known to be associated with immune compromise), studying common diseases in limited numbers (usually fewer than 100 diseases)[3–6]. Studies that systematically investigated diseases across clinical specialties, including those primarily managed and treated in primary care, are largely lacking, but are needed to understand why some patients with COVID-19 suffer from a severe outcome or dead, albeit at supposedly low-risk. Such a systematic, 'diseasome'-wide study can further improve our understanding of how variation in the host genome[7,8] confers risk for severe COVID-19 and guide drug target prioritisation strategies.

Here, we collate millions of health records from primary care, hospitalisations and cancer registrations, and death records among ~500,000 participants of the UK Biobank (UKB) into medical diagnosis concept terms[9], so-called 'phecodes'[10], to systematically assess the risk for severe COVID-19 and its post-acute sequalae, Long COVID, across the breadth of medical diagnosis. Apart from well-recognized high-risk patient groups, such as those with chronic kidney disease or those with compromised immune function, we demonstrate consistent evidence for the possible role of less recognized diseases and symptoms, including malaise and fatigue, based on survival and genetic susceptibility analyses. We finally observe that some genomic regions conferring a higher risk for severe COVID-19 might be protective for diseases that partially share pathomechanisms with COVID-19, or vice versa, with possible implications for drug development programs, such as TYK2-inhibitors that may increase the risk for severe COVID-19.

## Methods
### Study population
UKB is a prospective cohort study from the UK, which contains more than 500,000 volunteers between 40 and 69 years of age at inclusion. The study design, sample characteristics and genome-wide genotype data have been described in Sudlow et al.[9] and Bycroft et al.[11]. The UKB was approved by the National Research Ethics Service Committee Northwest Multi-Centre Haydock and all study procedures were performed in accordance with the World Medical Association Declaration of Helsinki ethical principles for medical research. All participants gave broad consent to use of their anonymised data and samples for any health-related research and for UKB to access their health-related records. UKB is registered as a Research Tissue Bank (https://www.hra.nhs.uk/planning-and-improving-research/policies-standards-legislation/research-tissue-banks-and-research-databases/) and hence all approved data applications (here ID: 44448) can use this ethical clearance to conduct their research. We included 502,460 individuals who had not withdrawn their consent. For survival analysis we considered a set of 438,917 individuals who were still alive at the beginning of the COVID-19 pandemic (01/01/2020) and had genetically inferred ancestry also beyond white Europeans. We chose the entire set of white Europeans ($n = 441,671$) that passed standard quality control for genetic analysis to maximise statistical power.

### COVID-19 and Long COVID outcome definitions
We defined a total of four different COVID-19 related outcomes closely aligned with previous studies[8,12,13]. We used hospital episode statistics to identify participants who had been 'hospitalised' with COVID-19 based on ICD-10 codes U07.1 and U07.2, and the same ICD-10 codes to identify participants who have died from/with COVID-19 based on death registries. We did not require a positive PCR COVID-19 test due to differences in local reporting of test results. We adopted a slightly more sophisticated definition for 'severe respiratory failure', demanding a positive COVID-19 test (based on test results released for England, Scotland, and Wales provided by UKB through the COVID-19 Second Generation Surveillance System) within a

month of acute respiratory failure, defined by ICD-10 codes J80, J96.00, J96.09, Z99.1 from hospital episode statistics or E85.1 and E85.2 when admitted to the intensive care unit. To define 'Long COVID' we used primary care data released by UKB (covid19_emis_gp_clinical.txt, covid19_tpp_gp_clinical.txt) searching for codes indicating suspected diagnosis [CTV3: Y2b89 – "Referral to post-COVID assessment clinic", Y2b8a – "Referral to Your COVID Recovery rehabilitation platform", Y2b87 – "Post-COVID-19 syndrome", and Y2b88 – "Signposting to Your COVID Recovery"; SNOMED-CT: 1325161000000102 – "Post-COVID-19 syndrome", 1325031000000108 – "Referral to post-COVID assessment clinic", 1325041000000104 – "Newcastle post-COVID syndrome Follow-up Screening Questionnaire", 1325181000000106 – "Referral to Your COVID Recovery rehabilitation platform", 1325021000000106 – "Ongoing symptomatic disease caused by severe acute respiratory syndrome coronavirus 2", 1325141000000103 – "Signposting to Your COVID Recovery", 1325081000000107 – "Assessment using Post-COVID-19 Functional Status Scale structured interview", 1325061000000103 – "Assessment using COVID-19 Yorkshire Rehabilitation Screening tool", 1325071000000105 – "Assessment using Newcastle post-COVID syndrome Follow-up Screening Questionnaire", 1325051000000101 – "COVID-19 Yorkshire Rehabilitation Screening tool"]. For each event, we took the earliest record to define disease onset.

We identified a total of 7507 (hospitalisation), 662 (respiratory failure), and 1546 cases (death), with first cases occurring end of January 2020. Due to restricted availability of primary care data, we only included records up until 30/09/2021 to identify 470 cases of Long COVID.

### Disease ascertainment
We collated electronic health records (EHRs) from primary and secondary care, cancer registries, and death certificates based on tables provided by UKB (gp_clinical.txt, covid19_emis_gp_clinical.txt, covid19_tpp_gp_clinical.txt, hesin_diag.txt, death.txt) downloaded in June 2022. We parsed all records to exclude codes with a recorded date before or within the year of birth of the participant to minimize coding errors from EHRs. We used mappings provided by UK Biobank to include self-reported conditions based on ICD-10 codes. For each data set separately, we generated mapping tables that link ICD-10, ICD-9, Read version 2, Clinical Terms Version 3 (CTV3) terms, or SNOMED-CT codes to a set of 1560 summarized clinical entities called phecodes[14,15] (Supplementary Data 1). For example, more than 90 ICD-10 codes can indicate participants with type 1 diabetes that are here collectively summarized under the phecode 'type 1 diabetes'[16]. We subsequently fused all data sources based on a common set of phecodes and retained for each participant and each phecode only the earliest entry across all EHR resources. We identified a total of 1448 phecodes with at least 100 cases in the overall UKB sample. For each participant and phecode, we kept only the earliest date as an indicator for disease onset and defined all events occurring before 01/01/2020 as prevalent, while we considered any event for genetic analysis. To increase the accessibility of our results, we use the term 'disease' instead of 'phecode' throughout the paper.

### Survival analysis
We used Cox-proportional hazard models to estimate the risk associated with each disease and any of the four COVID-19 related outcomes with age as the underlying scale, adjusting for sex (omitted for sex-specific diseases) and genetically inferred ancestry. For each COVID-19 outcome, we defined controls separately as all those participants without a corresponding record during the time course of the study. We repeated Cox-proportional hazard models considering all-cause death as a competing event rather than censoring as a sensitivity analysis. We selected 01/03/2020 as the starting point of our study and used 31/12/2022 (COVID-19 endpoints) or 30/09/2021 (Long COVID) as endpoints of the observation period depending on the availability of health record linkage. We computed Schoenfeld residuals to test for the proportional hazard assumption, and further computed time varying effects of diseases by introducing 6 months breaks. For each disease – COVID-19 model, we

considered all participants that passed inclusion criteria. We applied stringent multiple testing correction ($p < 0.05/4*1448 = 4.8 \times 10^{-8}$) and further filtered results for those possibly violating the proportional hazard assumption ($p < 10^{-3}$). To establish endpoint-specific associations, we performed meta-analysis across disease associations for all three COVID-19 endpoints derived using the R package *metafor (v.3.8.1)*. We performed additional sensitivity analysis using an extended set of confounders similar to previous work[17], including self-reported smoking status and alcohol consumption, body mass index, and Townsend deprivation index (all based on baseline values), healthcare utilization in the five years before the pandemic (number of stays and total days in hospital), as well as a variable indicating participants with two or more long-term conditions.

We tested for a potential modifying effect of sex, non-European ancestry, age ($\leq 65$ years vs $> 65$ years), and social deprivation (Townsend index above median vs below median; median = $-2.22$) on the results by systematically performing interaction testing, i.e., introducing a disease – sex/non-European ancestry interaction term into Cox-models. For the latter, we requested to have at least 50 observations in each group to ensure model convergence. We subsequently corrected for a total of 13,728 tests ($p < 3.6 \times 10^{-6}$). All statistical analysis were implemented using R v4.1.2.

### Disease network

We computed a sex-aware disease network using partial correlations as implemented in the R package *ppcor* (v.2.1.1) following previous work[18]. Briefly, partial correlations ($r_P$) account for the fact, that a correlation, or co-occurrence, between two diseases might be driven by a third or any other disease considered. We retained only partial correlations passing stringent multiple testing ($p < 4.9 \times 10^{-8}$) and $r_P > 0.02$ as we reasoned that a disease-disease network likely exhibits scale-free properties[19] with node degrees following a power law. The latter step omitted many significant, but very weak and potentially artificial edges. The final network contained 5212 edges connecting 1381 diseases. We then performed community detection based on the Girvan-Newman algorithm to identify groups of diseases that were more closely connected with each other compared to all other diseases in the network. We finally computed different node characteristics to identify diseases with important roles in the network. We implemented and visualized this analysis with the R package *igraph* (v.1.3.1).

### Genotyping, quality control, and participant selection

Details on genotyping for UKB have been reported in detail by Bycroft et al.[11]. Briefly, we used data from the 'v3' release of UKB containing the full set of Haplotype Reference Consortium (HRC) and 1000 Genomes imputed variants. We applied recommended sample exclusions by UKB including low quality control values, sex mismatch, and heterozygosity outliers. We defined a subset of 'white European' ancestry by clustering participants based on the first four genetic principal components derived from the genotyped data using a k-means clustering approach with $k = 5$. We classified all participants who belonged to the largest cluster and self-identified as of being 'white,' 'British', 'Any other white background', or 'Irish' as 'white European'. After application of quality control criteria and dropping participants who have withdrawn their consent, a total of 441,671 UKB participants were available for analysis with genotype and phenotype data.

We used only called or imputed genotypes and short insertions/deletions (here commonly referred to as single nucleotide polymorphisms (SNPs) for simplicity) with a minor allele frequency (MAF) > 0.001%, imputation score > 0.4 for common (MAF $\geq$ 0.5%) and > 0.9 for rare (MAF < 0.5%), within Hardy-Weinberg equilibrium ($p_{HWE} > 10^{-15}$), and minor allele count (MAC) > 10. This left us with 15,519,342 autosomal and X-chromosomal variants for statistical analysis. GRCh37 was used as reference genome assembly.

### Genome-wide association studies

We performed genome-wide association studies (GWAS) for a total of 1445 diseases with at least 80 cases ($n > 100$ prior genetic exclusions; 3 diseases dropped out) using REGENIE v2.2.4 via a two-step procedure to account for population structure as described in detail elsewhere[20]. We used a set of high-quality genotyped variants (MAF > 1%, MAC > 100, missingness < 10%, $p_{HWE} > 10^{-15}$) in the first step for individual trait predictions using the leave one chromosome out (LOCO) scheme. These predictions were used in the second step as offset to run logistic regression models with saddle point approximation to account for case/control imbalance and rare variant associations. Each model was adjusted for age, sex, genotyping batch, assessment centre, and the first ten genetic principal components. For diseases reported in only one sex ($n = 113$ in women, $n = 26$ in men), we excluded the respective sex from GWAS to avoid inflation by inappropriate controls. In general, we included all participant with a disease in their records as case and treated all other participants as controls to make best use of the computational efficacy of REGENIE. Testing for reported SNPs showed highly consistent results whether related diseases were included as controls rather than omitted. We used LD-score regression to test for genomic inflation (LDSC v1.0.1)[21].

### COVID-19 genetic correlation and Mendelian randomization

We downloaded GWAS summary statistics for two different endpoints related to COVID-19 (A2 – critical illness; B2 – hospitalisation) and Long COVID (stringent case definition vs broad control set) provided by the COVID-19 Host Genetics Initiative (release 7)[8,13]. We used summary statistics excluding UKB to avoid sample overlap. We computed genetic correlations as implemented by LD-score regression (LDSC v1.0.1)[21] with precomputed LD-scores, excluding the extended MHC region. To test for potentially causal associations of diseases onto COVID-19, we used genetic instruments identified in the present study for a total of 41 diseases with at least five genetic variants and evidence for significant genetic correlations in a two-sample MR setting. We used MR-PRESSO[22] as a first line tool as previously suggested[23] to account for possible pleiotropy and subsequently report effect estimates from inverse-variance weighted analysis as the primary results. We flagged MR results that showed signs of heterogeneity across instruments using Cochran Q statistic. We excluded any variants mapping to the MHC regions for all analysis and implemented MR using the R packages *MendelianRandomization* (v0.6.0)[24] and *TwoSampleMR* (v0.5.6)[25].

### Colocalisation at COVID-19 risk loci

We collected association statistics for a total of 49 independent risk loci for COVID-19 (selected based on regional clumping ( $\pm$ 500 kb) of COVID-19 HGI GWAS statistics excluding UKB participants, but SNPs available among imputed genetic data in UKB) across all 1445 diseases included in the genetic analysis. For variant – disease pairings passing a moderate significance threshold ($p < 10^{-6}$), we implemented statistical colocalization[26] accounting for multiple causal genetic variants via fine-mapping[27] using the R packages *coloc* (v.5.3.2) and *susieR* (v.0.11.92). We allowed for a maximum of five causal variants during fine-mapping of the disease and linked COVID-19 outcome (via a potentially shared genetic variant) and subsequently tested each credible set for colocalization. We applied a stringent prior to consider a shared signal ($p_{12} = 5 \times 10^{-6}$) and further filtered signals with evidence that the lead signal ($r^2$ with best remaining signal > 0.8) for COVID-19 was dropped from the set of overlapping genetic variants between our UKB GWAS and the COVID-19 GWAS.

### Reporting summary

Further information on research design is available in the Nature Portfolio Reporting Summary linked to this article.

### Results

Here, we systematically investigate the risk conferred by the presence and potential causal relevance of 1448 diseases for COVID-19 severity (hospitalisation, severe respiratory failure, and death) and Long COVID (Fig. 1), based on medical disorder concepts[14,16] defined and collated from >12 million medical records from primary (general practice), secondary care

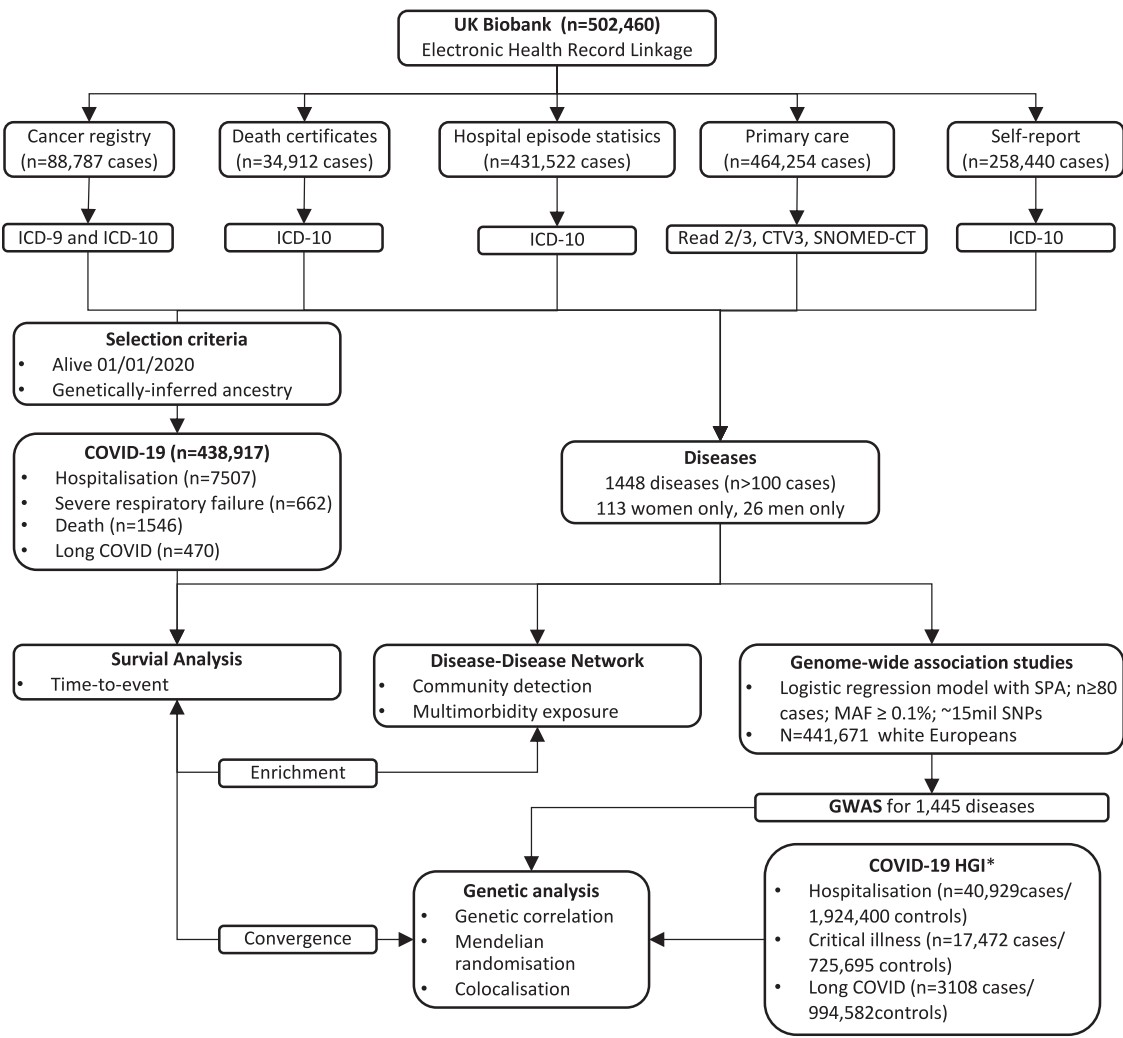

**Fig. 1 | Outline of the study design.** Scheme of the study design and analysis done, illustrating our workflow to define disease mechanisms that may causally contribute to severe COVID-19 or Long COVID. SNPs Single nucleotide polymorphisms; SPA = saddle point approximation; MAF = minor allele frequency; *COVID-19 HGI = COVID-19 Host Genetic Initiative, but excluding contributions from UK Biobank

(hospital admissions), and disease registry (cancer registry), death certificates, and patient-reported conditions among 502,460 UKB participants (Fig. 1 and Supplementary Data 1). Incorporating primary care data more than doubled case numbers for more than half (n = 817; 56.4%) of the diseases we considered (Supplementary Data 1).

### Disease risk profiles for COVID-19 and Long COVID

We identified 1128 significant ($p < 1.1 \times 10^{-5}$) disease – COVID-19 outcome associations, including almost half (n = 679) of the diseases considered with at least one of the four COVID-19 outcomes derived (Fig. 2 and Supplementary Data 2). Pre-existing diseases were almost exclusively associated with a higher risk for COVID-19 endpoints (median hazard ratio (HR): 2.39, range: 0.59–17.3), only two diseases (benign neoplasm of skin and varicella infection) were associated with a decreased risk. Associated diseases spanned almost all chapters of the ICD-10 (17 out of 18) but were consistently enriched in the chapters 'respiratory' (odds ratio [OR]: 5.96; $p$-value: $2.7 \times 10^{-8}$), 'circulatory' (OR: 2.95; $p$-value: $3.5 \times 10^{-7}$), and 'endocrine/metabolic' diseases (OR: 2.76; $p$-value: $9.1 \times 10^{-4}$) when associated with severe COVID-19. In contrast, pre-existing disease-codes classified as 'symptoms' were more than 13-fold enriched among diseases associated with an increased risk for Long COVID (OR: 13.2; $p$-value: $3.6 \times 10^{-8}$) but also hospitalisation (OR: 5.53; $p$-value: $9.9 \times 10^{-5}$) and death (OR: 3.06; $p$-value: $7.3 \times 10^{-3}$).

For COVID-19 requiring hospitalisation, we replicated and refined known associations with serious pre-existing diseases that have been previously used to identify clinically extremely vulnerable people. This included respiratory diseases like pseudomonal pneumonia (HR: 7.53, 95%-CI: 4.74–11.97; $p$-value $< 1.2 \times 10^{-17}$), acute renal failure (HR: 4.02, 95%-CI: 3.74–4.32, $p$-value: $<10^{-300}$) or type 2 diabetes with renal complications (HR: 7.44; 95%-CI: 5.67–9.76; $p$-value: $1.5 \times 10^{-47}$), as well as immune deficiencies (e.g., deficiency of humoral immunity HR: 6.02; 95%-CI: 4.36–8.31; $p$-value: $1.3 \times 10^{-27}$) or patients under immune suppression (e.g., liver transplants HR: 7.25 95%-CI: 4.51–11.68, $p$-value: $3.4 \times 10^{-16}$). However, we further observed strong associations with so far less recognized pre-existing mental health and psychiatric diseases and conditions with effect sizes comparable to those previously considered to identify extremely vulnerable people. This included symptoms of malaise and fatigue (HR: 2.17, 95%-CI: 2.07–2.27, $p$-value: $4.4 \times 10^{-222}$) or suicide attempts (HR 5.33, 95%-CI: 4.45–6.39, $p$-value: $3.6 \times 10^{-73}$). Most diseases (n = 641, 95.5%, $p_{hetero} > 10^{-3}$) associated with similar magnitude across all three different definitions of COVID-19 severity, with different forms of dementias ($p_{hetero} < 2.1 \times 10^{-24}$) being among the few exceptions, associating with hospitalisation (HR: 3.83; 95%-CI: 3.38–4.34; $p$-value: $2.3 \times 10^{-97}$) and death (HR: 10.82; 95%-CI: 9.15–12.80; $p$-value: $1.4 \times 10^{-170}$), but not severe respiratory failure (HR: 1.15; 95%-CI: 0.51–2.57; $p$-value: 0.74) due to COVID-19.

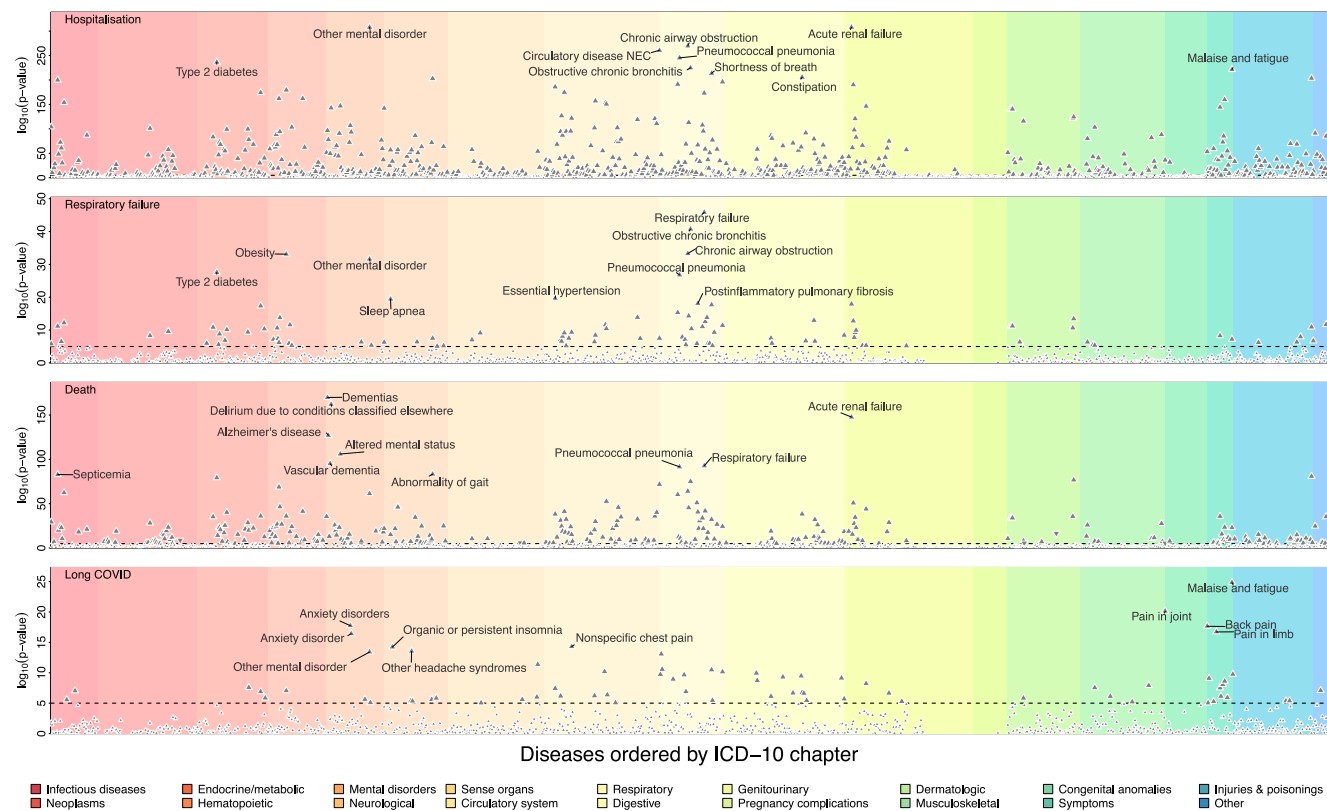

**Fig. 2 | Association results for three different COVID-19 outcomes and long COVID.** Each panel contains association statistics, *p*-values (triangles), from Cox-proportional hazard models (two-sided) testing for an association between the disease on the x-axis and three different COVID-19 outcomes, as well as Long COVID. Disease associations passing the multiple testing correction (dotted line, $p < 1.1 \times 10^{-5}$) are depicted by larger triangles of which facing up ones indicate positive, e.g., increased disease risk, associations and downward facing *vice versa*. The diseases are ordered by ICD-10 chapters (colours) and the top ten for each endpoint annotated. Underlying sample numbers and statistics can be found in Supplementary Data 1 and 2.

In contrast, pre-existing diseases associated with an increased risk for Long COVID only partially overlapped with those increasing the risk for severe COVID-19. Most notably, we replicated associations with anxiety disorders[28] (HR: 2.59; 95%-CI: 2.09–3.20; *p*-value:$1.8 \times 10^{-18}$) and other mental health symptoms, but most prominently with symptoms of malaise and fatigue (HR: 2.78; 95%-CI: 2.29–3.37; *p*-value:$1.5 \times 10^{-25}$) that are hallmarks of Long COVID and were also strongly associated with severe COVID-19.

Almost all significant associations (99.8%, n = 1126) were consistent when considering all-cause death as a competing event (Supplementary Data 3), and more than half (63.6%; n = 718) remained statistically significant ($p < 4.4 \times 10^{-5}$) when accounting for a large set of potential confounders in multivariable Cox-models (Supplementary Data 3). This suggests that potentially unreported associations, such as the increased risk for severe COVID-19 among patients reporting symptoms of malaise and fatigue (adjusted HR: 1.66, 95%-CI: 1.58 - 1.74, *p*-value = $7.3 \times 10^{-92}$), are not just a reflection of a general disease burden or other chronic diseases associated with a greater risk for severe COVID-19.

We observed limited evidence for effect modifications by sex (n = 7), non-European ancestry (n = 1), or age (n = 8), but not social deprivation, with 16 disease – COVID 19 pairings showing evidence of significant differences (Supplementary Data 4; p < $3.6 \times 10^{-6}$). All included stronger effects in women compared to men, e.g., gout for hospitalised COVID-19 (women: HR: 2.56, 95%-CI 2.21–2.96, *p*-value: $1.3 \times 10^{-36}$; men: HR: 1.46, 95%-CI: 1.34–1.58, *p*-value: $2.1 \times 10^{-19}$), among Europeans reporting vitamin D deficiencies (Europeans: HR: 2.31, 95%-CI: 2.13–2.51, *p*-value: $2.1 \times 10^{-87}$; non-Europeans: HR: 1.31, 95%-CI: 1.08–1.60, *p*-value = $5.5 \times 10^{-3}$), or among younger participants, e.g., disorders of magnesium metabolism and death with COVID-19 as a likely result of renal failure (age ≤ 65 years: HR:

42.98, 95%-CI: 20.10–91.90, *p*-value: $3.0 \times 10^{-22}$; age > 65 years: HR: 5.35, 95%-CI: 3.51–8.16, *p*-value: $5.9 \times 10^{-15}$).

## Complex patterns of multimorbidity are associated with increased risk

We next derived a disease-disease network[18] (Fig. 3a) to understand, whether the large set of diseases associated with an increased risk for severe COVID-19 act independently or rather reflect an increased risk among participants suffering from multiple pre-existing conditions, i.e., multimorbidity. The network contained a total of 1381 diseases connected through 5212 edges based on non-random co-occurrence (Supplementary Data 5a, b). Diseases segregated into 31 'communities' being more strongly connected to each other compared to the rest of the network (Fig. 3b, c).

Two disease communities were consistently and strongly enriched for diseases associated with severe COVID-19. The first (e.g., OR: 5.20; *p*-value = $2.2 \times 10^{-10}$; for severe respiratory failure) community was strongly enriched for circulatory (OR: 17.6; *p*-value = $4.4 \times 10^{-39}$) and respiratory (OR: 10.3; *p*-value: $7.8 \times 10^{-16}$) diseases, closely resembling the cardio-respiratory risk profile already described above (Fig. 3b). The second community consisted of diverse endocrine (OR: 6.19; *p*-value: $1.9 \times 10^{-13}$) and circulatory disease (OR: 3.75; *p*-value: $5.4 \times 10^{-8}$), and largely reflected the renal-diabetic risk profile (Fig. 3c). Accordingly, for each disease acquired during lifetime within the latter disease community, participants' risk increased by 18% and 20% to be hospitalised (HR: 1.18; 95%-CI: 1.17–1.18; *p*-value: $p < 10^{-300}$) or die with COVID-19 (HR: 1.20; 1.19–1.20; *p*-value < $10^{-300}$), respectively.

Diseases increasing the risk for severe COVID-19, but not Long COVID further significantly correlated with hub status (e.g., hospitalisation: r = 0.59; *p*-value: $2.8 \times 10^{-124}$) in the disease-disease network (Fig. 3d), that is,

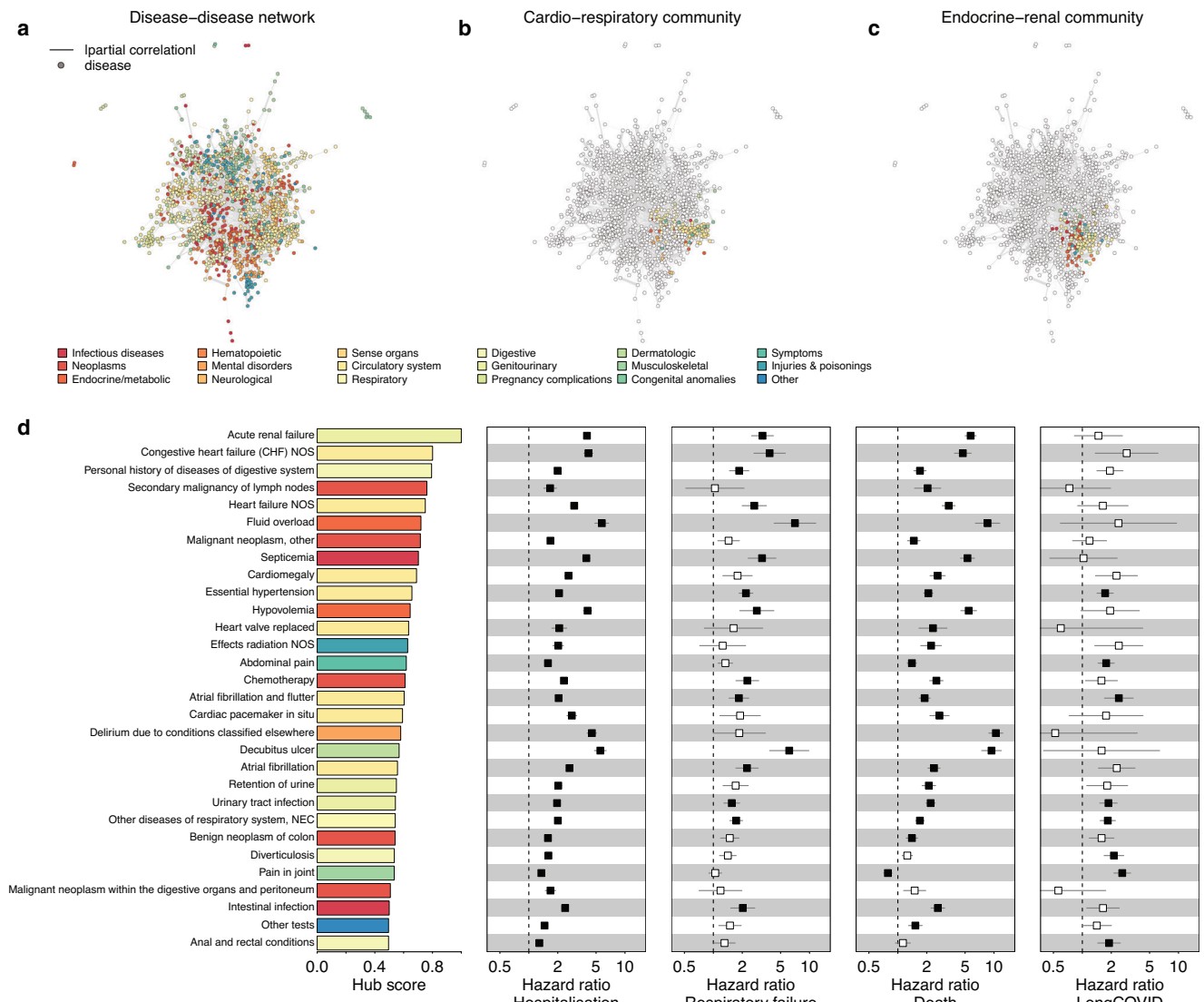

**Fig. 3 | Disease-disease network and hub score. a** Disease – disease network based on significant ($p < 4.8 \times 10^{-8}$) positive partial correlations (two-sided). Nodes (diseases) are coloured by ICD-10 chapters and strength of partial correlation depicted by width of the edges. The underlying data can be found in Supplementary Data 5a–c Same network, but only highlighting two disease communities strongly enriched for associations with severe COVID-19. **d** Hub score for the 30 diseases with highest values and associated association statistics, hazard ratios (rectangle) with 95%-confidence intervals (lines), from Cox-proportional hazard models (two-sided). Significant associations are indicated by filled boxes. Colours according to ICD-10 chapters. All underlying data can be found in Supplementary Data 2 and 5b.

diseases that connect a large cluster of diseases to the rest of the network and might hence be considered as multimorbidity hotspots. For example, acute renal failure, strongly associated with severe COVID-19 (Fig. 3d), showed strong partial correlations with 30 other diseases and patients are hence prone to complex multimorbidity. However, the imperfect correlation between hub status and disease-association profiles indicates that certain forms of multimorbidity, such those related to secondary malignancies of lymph nodes, are possibly less related to severe COVID-19.

## Convergence of associated disease risk and genetic liability

We next systematically characterised whether diseases identified to be associated with COVID-19 severity or Long COVID shared genetic similarity with host genetic susceptibility to severe COVID-19 to understand potential underlying causal mechanisms. We computed genetic correlation estimates for all 1128 disease – COVID-19 outcome pairs and observed 75 pairs (6.6%) that showed evidence for significant ($p < 4.4 \times 10^{-5}$) and directionally consistent genetic correlations (Fig. 4 and Supplementary Data 6), indicating a putatively causal link of any of 57 unique diseases on severe COVID-19. We did not observe evidence of convergence for Long

COVID, which might likely be explained by the still low statistical power for the respective genome-wide association study[13].

The diseases with consistent evidence from survival and genetic analysis included well-described risk-increasing effects of pre-existing endocrine (e.g., type 2 diabetes), respiratory (e.g., respiratory failure), or renal (e.g., chronic kidney disease) diseases, but also digestive (e.g., gastritis and duodenitis), or musculoskeletal (e.g., rheumatoid arthritis) diseases, and further symptoms of malaise and fatigue ($r_G = 0.26$; $p$-value $= 4.7 \times 10^{-6}$) and abdominal pain ($r_G = 0.33$; $p = 2.5 \times 10^{-11}$), as well as adverse reactions to drugs (e.g., poisoning by antibiotics: $r_G = 0.38$; $p$-value $= 2.2 \times 10^{-6}$). Findings that collectively demonstrated the need for a comprehensive assessment of disease-risk beyond few, selected common chronic conditions.

Among the 41 diseases for which we had sufficient genetic instruments to perform more stringent Mendelian randomization (MR) analyses to assess causality, we observed only nominally significant ($p < 0.05$) evidence for gout and hospitalisation (OR: 1.03; 95%-CI: 1.01–1.05, $p$-value: 0.03), as well as arthropathy not elsewhere specified (OR: 1.28; 95%-CI: 1.06–1.55; $p$-value: 0.02) and unspecified monoarthrtitis (OR: 1.21; 95%-CI: 1.04–1.41; $p$-value: 0.02) for severe COVID-19 (Supplementary Data 7). While we

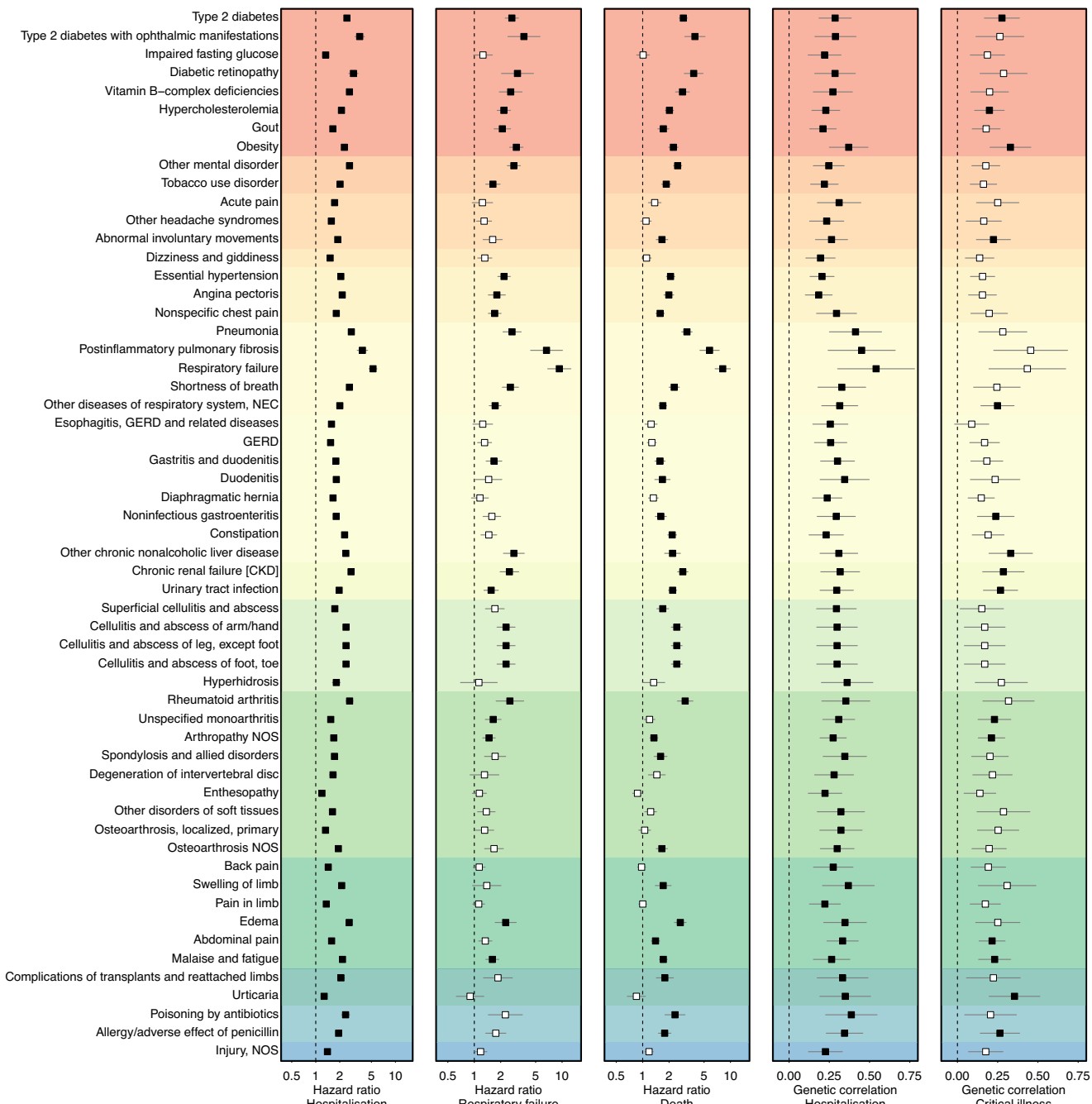

**Fig. 4 | Convergence of Cox-models and genetic correlations.** The first three panels show association statistics, hazard ratios (rectangle) and 95%-confidence interval (lines), for 57 diseases with evidence of convergence with genetic correlation analysis, that are shown in the last two panels (rectangle – genetic correlation; lines – 95%-confidence intervals). Disease have been grouped by ICD-10 chapters and coloured accordingly (see Figs. 2 or 3 for legend). NOS = not elsewhere specified; All underlying data can be found in Supplementary Data 1 (sample numbers), 2 and 6.

might have been still underpowered for many diseases, this leaves the possibility that convergence of survival and genetic correlation analysis might, in part, be explained by shared risk factors.

**Evidence for partially opposing roles of shared molecular mechanisms between severe COVID-19 and related disorders**

To finally understand possible molecular mechanisms linking the 'diseasome' to COVID-19, we systematically profiled disease associations across 49 independent genomic regions linked to COVID-19 or Long COVID. We observed strong and robust evidence of a genetic signal shared between severe COVID-19 and a total of 33 diseases at nine loci (posterior probability (PP) > 80%) (Fig. 5a and Supplementary Data 8). Apart from known

pleiotropic loci, such as *ABO* and *FUT2* coding for blood group types, this included respiratory risk loci, albeit with contradicting effect estimates for three loci (Fig. 5b). While COVID-19 risk increasing alleles at *LZTFL1* and *TRIM4* were consistently associated with a higher risk for viral pneumonia and post-inflammatory pulmonary fibrosis, respectively, risk-increasing alleles at *MUC5B*, *NPNT*, and *PSMD3* were inversely associated with post-inflammatory pulmonary fibrosis and asthma. An observation that extended even beyond shared loci (Fig. 5c) illustrating a general trend of phenotypic divergence of genetic effects on diseases that share pathological features with severe COVID-19.

A notable observation was the *TYK2* locus that has previously been suggested to indicate the efficacy of successfully repurposed drugs for severe

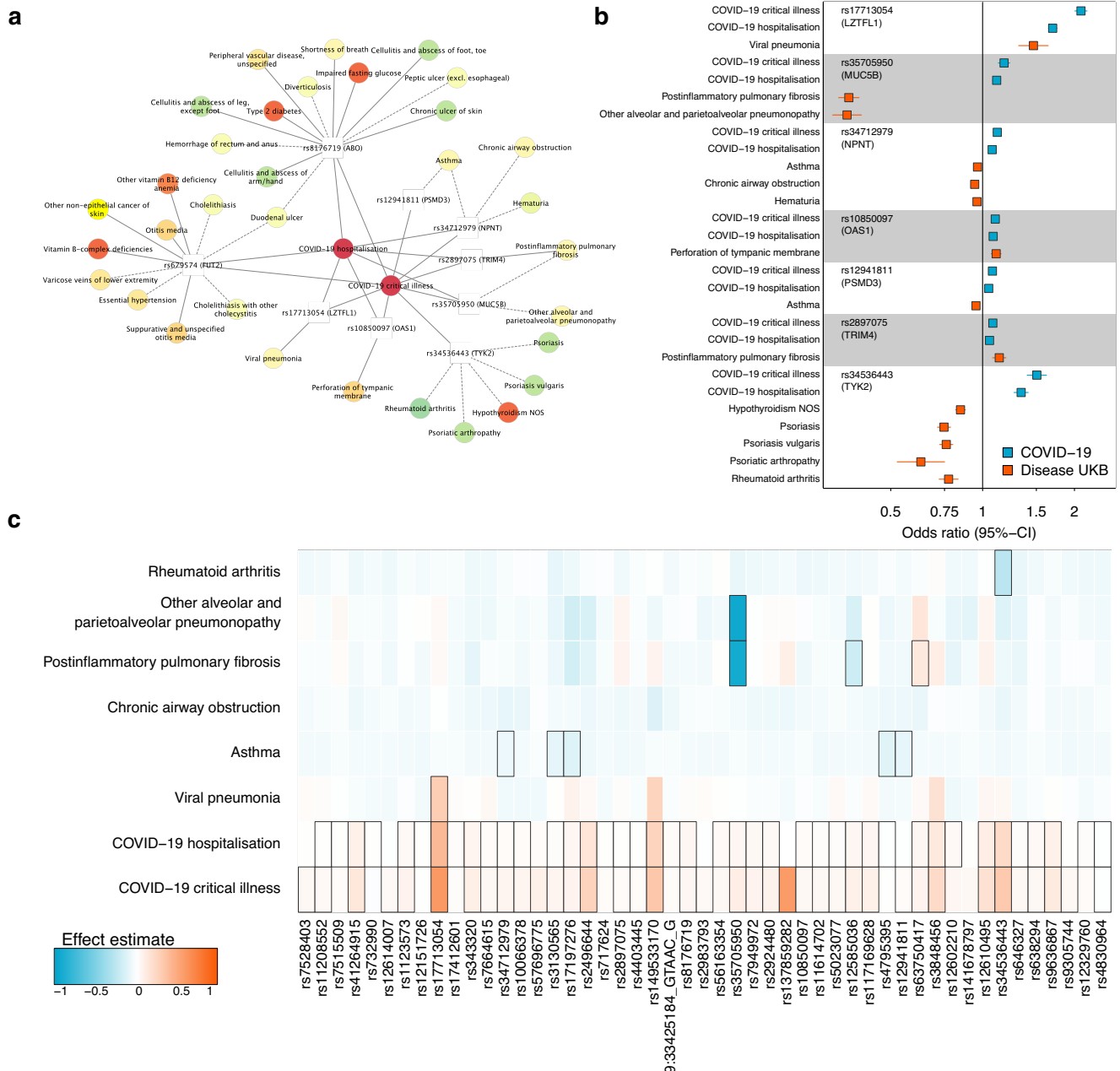

**Fig. 5 | Shared genetic architecture at COVID-19 risk loci. a** Network representation of significant (PP > 80%) colocalization results. Loci are depicted as white rectangles and diseases as coloured nodes according to ICD-10 chapters. Edges represent strong evidence for colocalization, and solid lines indicate a risk-increasing effect of the COVID-19 risk increasing allele, whereas dashed lines indicate protective effects. Underlying data can be found in Supplementary Data 8. **b** Forest plot displaying hazard ratios (rectangle) with 95%-confidence intervals (lines) for each variant and different COVID-19 and colocalising disease outcomes. Effect estimates for COVID-19 have been obtained from the COVID-19 Host Genetic Initiative and effect estimates for diseases in the present study. All estimates are derived from logistic regression models. **c** Heatmap of effect estimates across 49 independent genetic loci associated with increased risk for sever COVID-19 and corresponding effects on six selected traits that showed evidence of colocalization at least one other locus. Black rectangles indicate genome-wide significant effects ($p < 5 \times 10^{-8}$). NOS not elsewhere specified; All underlying data can be found in Supplementary Data 1 and 8 or is given in the data availability statement.

COVID-19[29]. Briefly, *TYK2* encodes for tyrosine kinase 2 (TYK2) a protein partially targeted by Janus kinase (JAK) inhibitors like baricitinib, that have been approved for rheumatoid arthritis and successfully repurposed for severe COVID-19, although predating possible evidence from genetic studies[30–32]. Accordingly, we observed that the same genetic variant, rs34536443 (PP = 99.8%), associated with the risk for severe COVID-19 was also associated with, amongst others, the risk of rheumatoid arthritis, but in opposing effect directions (Fig. 5b). Rs34536443 is a loss-of-function missense variant (p.Pro1104Ala) for TYK2 and the functionally impairing minor C allele was associated with a 50% increased risk for

severe COVID-19 (odds ratio: 1.50; 95%-CI: 1.40– 1.62, *p*-value = 4.3 x 10$^{-29}$) but a 23% reduced risk for rheumatoid arthritis (odds ratio: 0.77; 95%-CI: 0.72–0.83; *p*-value = 2.4 x 10$^{-12}$) as well as other autoimmune diseases, in particular psoriasis (Supplementary Data 8). While the discrepancy between the success of the drug and genetic inference might be explained by the rather weak affinity of baricitinib for TYK2[33], patients undergoing trials with TYK2-inhibitors for psoriasis[34] might be at an elevated risk for severe COVID-19. This observation seemingly aligns with studies on *Tyk2*$^{-/-}$ mouse models reporting an impaired immune response to viral infections[35].

## Discussion

An immediate understanding which patients are at greatest risk for severe COVID-19 and possibly death has proven to be instrumental to triage patients early in the pandemic to allocate critical care resources, such as ventilation or extracorporeal membrane oxygenation and, later, vaccination as well. The vast majority of studies[3–6], however, focussed on a rather narrow set of common, usually chronic, conditions in the risk assessment leaving a considerable number of severe COVID-19 cases unexplained. We demonstrate here how capitalizing on the whole breadth of medical diagnoses through electronic health record linkage revealed 1) so far largely neglected patient populations at considerable risk, including those reporting symptoms of malaise and fatigue, and 2) that patients with multiple pre-existing conditions, in particular cardio-respiratory and endocrine-renal diseases, are probably at highest risk. Via integration of host genetics, we further provide evidence that a considerable set of diverse diseases may causally drive, or at least share causal drivers with, the risk for severe COVID-19, and exemplify how disease-wide characterisation of specific risk loci can inform disease mechanism and derivation of potentially druggable targets or adverse effects.

Among the diseases for which we observed consistent evidence from survival and genetic analysis to be linked to severe COVID-19 were multiple examples that have been rarely if at all reported. For example, we observed consistent evidence that symptoms of malaise and fatigue, as well as chronic fatigue, predispose to severe COVID-19. While the vast amount of literature currently discusses or reported these symptoms and disease as characteristics for COVID-19 and its post-acute sequelae[28,36], little to nothing is known why patients reporting fatigue might be at higher risk. While our definition of 'malaise and fatigue' covered a broad range of partially unspecific medical codes with most cases (n = 83,316 out of 87,908, 92.4%) originating from primary care, we observed consistent evidence for the refined diagnosis of chronic fatigue classified as post-viral fatigue symptom (Supplementary Data 2). A hypothesis might be, that patients that are already suffering from post-viral symptoms are at a greater risk in general to suffer from more severe courses of viral infections through yet to be identified mechanisms, that may well comprise an altered immune response. However, the evidence we provide does not preclude the existence of general, currently inaccessible, risk factors that predispose to more severe long-term consequences of viral infections.

Our extensive genetic analysis revealed some partially contradicting findings that may point to a segregation of overall genetic susceptibility and risk conferred by specific loci and mechanisms, replicating and augmenting findings from a previous study in the Million Veterans Study[37]. For example, we observed consistent evidence that pre-existing post-inflammatory pulmonary fibrosis, likely representing cases of idiopathic pulmonary fibrosis, is a strong risk factor for severe COVID-19 and death, and genome-wide effects were highly correlated between both ($r_G$=0.45, $p$ = 2.3 x 10$^{-5}$), but effects at one of the strongest risk loci for post-inflammatory pulmonary fibrosis were protective for severe COVID-19. Our results thereby extend previous observations of misaligning effects at the *MUC5B* locus and idiopathic pulmonary fibrosis[38,39]. Results that might be explained by a latent, genome-wide risk component (as genome-wide significant loci do not contribute to genetic correlation analysis) that predisposes to severe lung fibrosis irrespective of the exact trigger, and specific molecular pathways characteristic for each disease that differ based on the required immune response to combat the infection. Cell-type and state-specific effects of shared genetic variants or possible design artefacts of GWAS studies of infectious disease, by which certain patient groups are 'underrepresented' due to tailored shielding efforts to minimize viral exposure, are other possible explanations. A similar paradoxical effect at the *TYK2* locus highlights the unique potential of integrating electronic health care records with genetic data to guide drug target identification and risk estimation, including emerging diseases and targets in clinical trials.

There are a number of limitations that need to be taken in consideration when interpreting our results. Firstly, the COVID-19 pandemic was characterised by strong disruptions of social life and health care, with different waves of new SARS-CoV-2 variants of different pathogenicity, lockdowns, and implementation of vaccines programs, all of which will have influenced the general risk to develop severe COVID-19 for which we could not control for in survival analysis. However, we observed generally little evidence of violation of the proportional hazard assumptions and filtered associations with evidence for strong violations. Secondly, we cannot exclude the possibility that the multitude of diseases associated with severe COVID-19 might also be explained by shared, generic risk factors, such as obesity or smoking, and we implemented sensitivity analysis and comprehensive genetic analysis to mitigate possible confounding, although even larger genetic studies are needed to identify robust genetic signals for diseases like chronic fatigue and other rare diseases that we linked to COVID-19. Thirdly, while we obtained little evidence that disease-risk patterns differ across ancestries, the UK Biobank cohort is not a representative sample of the general population and does not sufficiently cover underrepresented populations, e.g., ethnic minorities, and additional work is needed to verify our observations in other populations. Lastly, while our effort to collate and harmonize electronic health records across various sources into medical concept terms covered almost 1500 diseases, it is still only an approximation of the complexity of medical diagnosis and more work, using electronic health records at a national scale, is needed to refine and augment the space of diseases to investigate.

Our results demonstrate the unique potential of integrating health records from primary and secondary care with host genetic data to 1) rapidly identify patients at highest risk beyond commonly assessed risk groups, 2) understand pathological pathways, and 3) inform druggable strategies for emerging health threats, such as COVID-19.

## Data availability

Genome-wide summary statistics for diseases ('phecodes') in UK Biobank were generated, in part, from primary care data released to UK Biobank specifically for the use of COVID-19 research only, according to COPI regulations, and can therefore not be made publicly available. Access to individual level data can be requested by bona fide researchers from the UK Biobank (https://www.ukbiobank.ac.uk/). This research has been conducted under the application 44448. Mapping of Read codes to phecodes can be downloaded from https://github.com/spiros/ukbiobank-read-to-phecode. We downloaded GWAS summary statistics for two different endpoints related to COVID-19 (A2 – critical illness; B2 – hospitalisation) and Long COVID (stringent case definition vs broad control set) provided by the COVID-19 Host Genetics Initiative (release 7) from https://www.covid19hg.org/. Source data for the figures are available in Supplementary Data 1-8.

## Code availability

Associated code and scripts for the analysis can be found here https://github.com/comp-med/phecode-covid19-ukb[40].

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

## Acknowledgements

The authors acknowledge the Scientific Computing of the IT Division at the Charité - Universitätsmedizin Berlin for providing computational resources that have contributed to the research results reported in this paper (https://www.charite.de/en/research/research_support_services/research_infrastructure/science_it/#c30646061). This work was supported by funding of the German Centre for Cardiovascular Research (DZHK) and the German Ministry of Education and Research (BMBF), and the UKRI/NIHR Strategic Priorities Award in Multimorbidity Research for the Multimorbidity Mechanism and Therapeutics Research Collaborative (MR/V033867/1) to C.L.. H.H. and S.D. are supported by Health Data Research UK and the National Institute for Health Research (NIHR) Biomedical Research Centre at University College London NHS Hospitals Trust. M.A. and G.K. are supported by National Institutes of Health/National Institute on Aging grants RF1AG059093, U01AG061359, U19AG063744, R01AG069901, U19AG074879, and R01AG081322. G.K. also received funding from the German Federal Ministry of Education and Research (BMBF) (BiomarKid, 01EA2203B) under the umbrella of the European Joint Programming Initiative "A Healthy Diet for a Healthy Life" (JPI HDHL) and of the ERA-NET Cofund ERA-HDHL (GA N° 696295 of the EU Horizon 2020 Research and Innovation Programme) and of the German Network for Mitochondrial Disorders (mitoNET, 01GM1906C). This work was supported by the de.NBI Cloud within the German Network for Bioinformatics Infrastructure (de.NBI) funded by the German Federal Ministry of Education and Research (BMBF) (031A532B, 031A533A, 031A533B, 031A534A, 031A535A, 031A537A, 031A537B, 031A537C, 031A537D, 031A538A).

## Author contributions

Conceptualization: M.P., H.H., C.L. Data curation/Software: M.P., S.D., S.Y., M.U., M.A. Formal Analysis: M.P., S.Y. Methodology: M.P., S.D., T.N. Visualization: M.P. Funding acquisition: C.L., H.H. Project administration: C.L., H.H. Supervision: M.P., C.L., H.H. Writing – original draft: M.P., C.L., H.H. Writing – review & editing: S.D., S.Y., M.A., G.K., T.N.

## Competing interests

The authors declare no competing interests.
