## [Peer Review File · Communications Medicine]

Reviewers' comments:

Reviewer #1 (Remarks to the Author):

The study of Pietzner et al aimed at exploring a breadth of medical diagnoses, including common, non-fatal diseases, to improve understanding on risk factors and pre-existing conditions predisposing to COVID-19 severity and Long COVID. Inputs for analyses included >12 million primary care and hospitalisation health records from ~500,000 UK Biobank participants into 1448 collated disease terms. Authors systematically screened for diseases predisposing to severe COVID-19 (defined as requiring hospitalisation or death) and Long COVID. Authors found that diseases increasing risk for severe COVID-19 spanned almost all clinical specialities and were enriched in clusters of cardio-respiratory and endocrine-renal diseases. Further survival and Mendelian Randomization analyses showed consistency with some of the observational findings, including symptoms of malaise and fatigue. Authors concluded that so far largely neglected patient populations are at considerable risk, including those reporting symptoms of malaise and fatigue, and that patients with multiple pre-existing conditions, in particular cardio-respiratory and endocrine-renal diseases, are probably at highest risk.

This is a very interesting and impressive comprehensive analyses on risk factors and pre-existing conditions that predispose to COVID-19 severity and Long COVID.

The manuscript is extremely well-written; methods applied for the analyses have been well described and seemed appropriate. Results have been clearly presented.

I only have few suggestions to further improve clarity of the method sections, in particular for the survival analyses:

In the survival analysis section, I suggest to clarify:

- whether competing risk approaches were used given the dependency between outcomes;
- whether there has been participant drop out before the censoring date and how these were included in the analyses
- Approaches using inverse probability of treatment weighting might have been used for adjustment of Hazard ratios as described elsewhere (Zang C et al, Nat Comm 2023)

Minor comment: Typo line 111; we should read 30/09/2021 instead of 30/09/2921

Reviewer #2 (Remarks to the Author):

In this manuscript the authors evaluate medical records for ~500k UK patients enrolled in the UK Biobank. They map the medical codes to 1,448 phecodes and perform several analyses. They evaluate endpoints of COVID-19 hospitalization, COVID-19 respiratory failure, COVID-19 death, and Long COVID. Its an exploratory analysis that looks for disease risk factors and the effects of certain genetic risk loci.

The scope of the work is very impressive as it includes both epidemiology and genetics results. While many of the findings have been previously reported, the authors do point out certain novel findings (ie fatigue and malaise as a risk factor, genetic risk loci having divergent associations with critical disease)

Major comments:

1. I do wonder if certain biases are being introduced based on how the authors have set up the initial epidemiology study for assessing risk factors:

- Can the authors justify the use of the Cox-proportional hazard ratio? It is introducing a temporal dimension which in this case, if the starting point is early 2020, could introduce a certain bias where those with increased risk of early infection are over-represented (ie those in nursing homes that succumb to infection the earliest are over represented in deaths)

- I would clarify and specify the controls for the reader. For each of the endpoints, is the model comparing those with the endpoint to those without that endpoint. If so can the authors specify this, and potentially include a discussion of the limitations for this type of control group. For example, when I look at the diseases associated with death these to me seem like diseases that are risk factors for death from any cause. Are the controls extremely low health care utilizers?

- The authors report evaluating effect modification by sex and non-European ancestry. Can they also report how age and healthcare utilization (and potentially deprivation index) also influence these associations. Increased age and healthcare utilization will lead to increased diagnoses codes, and could potentially explain many of the associations.

2. The finding of malaise and fatigue increased association is interesting. I'm not sure I agree with the authors hypothesis that this is evidence that previous post-viral infections pre-dispose you to increase risk of COVID-19 severity. It seems to me like malaise and fatigue could be associated with a wide-variety of diseases including psychological, neurological, cardiac diseases, etc. I wonder if they controlled for other disease types and utilization, whether this relationship would disappear? It still seems like an interesting hypothesis-generating finding, but as a reader I'm skeptical that this is anything different than known previous risk factors, but now reported as a symptom rather than a chronic disease. (I also would double check that these diagnoses are definitely from before 2020 and not during the pandemic)

3. As a reader I'm somewhat confused about the divergent effect of certain genetic loci and the authors explanation for this is that those carrying a high-risk allele may adhere to more cautious measures? If that was the case, shouldn't the diseases also show protective effects? For example, shouldn't we see

that the ICD codes for pulmonary fibrosis are also associated with protection? Can the authors further justify or explain this finding.

4. Some of your Results include topics that should be in the Discussion. For example, I would move Lines 246-250 and 263-267 to the discussion

Minor Point

1. Its interesting that the respiratory failure endpoint has less total patients than deaths due to COVID-19. While this is likely in part due to the need for a positive test and the diagnosis code for the resp failure, I also wonder if certain diagnosis codes were excluded that should have been included (for example, J96.01 – Acute respiratory failure with hypoxia, J96.02...). However, I am listing this as a minor comment, because I doubt their exclusion really effects the findings of the author's study, but I wanted to point it out for their future epidemiology studies.

2. In Results: Line 111, fix the date

3. Supplementary Table 2? Change to landscape and reformat. It it's current format it is very challenging to read.

4. Seems like supplementary table only has supplement table 2, although I see others in the excel file?

5. Line 114 - What are the two exceptions? Also can you clarify what you mean here? Is this of all ~1,400 diseases evaluated?

6. Line 400 , two a's

7. Line 235, 49 known genomic regions. Can you cite where this is coming from?

8. Line 408 can the author include definitions for Y2b89, Y2b8a, Y2b87, and Y2b88.

Reviewers' comments:

Reviewer #1 (Remarks to the Author):

The study of Pietzner et al aimed at exploring a breadth of medical diagnoses, including common, non-fatal diseases, to improve understanding on risk factors and pre-existing conditions predisposing to COVID-19 severity and Long COVID. Inputs for analyses included >12 million primary care and hospitalisation health records from ~500,000 UK Biobank participants into 1448 collated disease terms. Authors systematically screened for diseases predisposing to severe COVID-19 (defined as requiring hospitalisation or death) and Long COVID. Authors found that diseases increasing risk for severe COVID-19 spanned almost all clinical specialities and were enriched in clusters of cardio-respiratory and endocrine-renal diseases. Further survival and Mendelian Randomization analyses showed consistency with some of the observational findings, including symptoms of malaise and fatigue. Authors concluded that so far largely neglected patient populations are at considerable risk, including those reporting symptoms of malaise and fatigue, and that patients with multiple pre-existing conditions, in particular cardio-respiratory and endocrine-renal diseases, are probably at highest risk.

This is a very interesting and impressive comprehensive analyses on risk factors and pre-existing conditions that predispose to COVID-19 severity and Long COVID. The manuscript is extremely well-written; methods applied for the analyses have been well described and seemed appropriate. Results have been clearly presented.

I only have few suggestions to further improve clarity of the method sections, in particular for the survival analyses:

In the survival analysis section, I suggest to clarify:

- whether competing risk approaches were used given the dependency between outcomes;

R1.1 We have followed this helpful suggestion by the reviewer and now performed additional analysis considering all-cause death as a competing event in a multi-state Cox-proportional hazard model. These results were highly comparable to our original results (99.8% statistically significant), and we now present them as sensitivity analysis (p5, lines 193-200).

- whether there has been participant drop out before the censoring date and how these were included in the analyses

R1.2 We excluded 63,543 participants that either withdrew their consent (n=77), had died before that start of the pandemic (n=28,606), or who had no information on their genetically inferred ancestry (n=37,011). We did not apply any specific sensitivity analysis as missing genetic information was not related to any of the outcomes considered. However, we acknowledge, that the cohort considered represented a likely healthier sample compared to the general middle-age to old UK population at the time of the pandemic (p12, line 364-367).

- Approaches using inverse probability of treatment weighting might have been used for adjustment of Hazard ratios as described elsewhere (Zang C et al, Nat Comm 2023)

R1.3 We thank the reviewer for this important suggestion and now provide additional sensitivity analyses using multivariable Cox-proportional hazard models adjusting for additional confounders like those from Zang et al. used to compute IPTWs. A procedure statistically equivalent to IPTW and also suggested by reviewer 2 (see R2.1c). About two-thirds of the significant associations remained statistically significant after adjustment and results are now reported in the revised manuscript (p5, line 193-200).

Minor comment: Typo line 111; we should read 30/09/2021 instead of 30/09/2921

R1.4 This typo has now been corrected.

Reviewer #2 (Remarks to the Author):

In this manuscript the authors evaluate medical records for ~500k UK patients enrolled in the UK Biobank. They map the medical codes to 1,448 phecodes and perform several analyses. They evaluate endpoints of COVID-19 hospitalization, COVID-19 respiratory failure, COVID-19 death, and Long COVID. Its an exploratory analysis that looks for disease risk factors and the effects of certain genetic risk loci.

The scope of the work is very impressive as it includes both epidemiology and genetics results. While many of the findings have been previously reported, the authors do point out certain novel findings (ie fatigue and malaise as a risk factor, genetic risk loci having divergent associations with critical disease)

Major comments:

1. I do wonder if certain biases are being introduced based on how the authors have set up the initial epidemiology study for assessing risk factors:

- Can the authors justify the use of the Cox-proportional hazard ratio? It is introducing a temporal dimension which in this case, if the starting point is early 2020, could introduce a certain bias where those with increased risk of early infection are over-represented (ie those in nursing homes that succumb to infection the earliest are over represented in deaths)

R2.1a We agree that proportional hazards is a central assumption for Cox models and we have implemented rigorous testing of the proportional hazard assumption and associated p-values are reported in Supplemental Table 2. We detected only a very small number of disease – COVID-19 associations that violated the proportional hazard assumptions, including a higher risk to be hospitalized with COVID-19 when diagnosed with dementia early (≤ 6 months; HR: 7.65, 95%-CI: 6.02-9.73, p-value= 6.3×10^{-62}) versus later (≥ 24 months; HR: 2.93, 95%-CI: 1.73-3.84, p-value= 2.9×10^{-6}) during the pandemic. A finding in line with the suspicion of the reviewer that diseases indicating particularly vulnerable groups will change over time. We now acknowledge this limitation in the manuscript (p11, lines 351-357).

- I would clarify and specify the controls for the reader. For each of the endpoints, is the model comparing those with the endpoint to those without that endpoint. If so can the authors

specify this, and potentially include a discussion of the limitations for this type of control group. For example, when I look at the diseases associated with death these to me seem like diseases that are risk factors for death from any cause. Are the controls extremely low health care utilizers?

R2.1b We now clarify the definition of controls more clearly in the revised version of the manuscript (p14, lines 441-445). Briefly, for each COVID-19 endpoint separately, we defined controls as those participants not having a related record. We further censored deaths. To further clarify, for each disease considered as exposure, we treated all participants with a related code in their medical records preceding the pandemic as a 'case' and everyone else as a 'control'. In doing so, controls were not enriched for low health care utilizers but rather represented an almost representative sample of the general population.

We also added novel sensitivity analysis treating all-cause mortality as a competing event in multi-state Cox-proportional hazard models that demonstrated an independent effect of the selected diseases on COVID-19 related deaths rather than all-cause mortality (p5, lines 193-200).

- The authors report evaluating effect modification by sex and non-European ancestry. Can they also report how age and healthcare utilization (and potentially deprivation index) also influence these associations. Increased age and healthcare utilization will lead to increased diagnoses codes, and could potentially explain many of the associations.

R2.1c We performed the following sensitivity analysis to address this important comment of the reviewer: 1) additional interaction analysis by age and deprivation groups, and 2) additionally adjusting of healthcare utilization in multivariable Cox models. We observed limited evidence for significant effect modifications by age (n=8) or deprivation strata (none; p6, lines 201210), and further almost two-thirds of the presented associations (63.6%; n=718) were still statistical significant after controlling for a large set of confounders, including health care utilization, lifestyle factors, obesity, social deprivation, and a multimorbidity index. This information has been added to the revised manuscript (p5, line 193-200).

2. The finding of malaise and fatigue increased association is interesting. I'm not sure I agree with the authors hypothesis that this is evidence that previous post-viral infections predispose you to increase risk of COVID-19 severity. It seems to me like malaise and fatigue could be associated with a wide-variety of diseases including psychological, neurological, cardiac diseases, etc. I wonder if they controlled for other disease types and utilization, whether this relationship would disappear? It still seems like an interesting hypothesis-generating finding, but as a reader I'm skeptical that this is anything different than known previous risk factors, but now reported as a symptom rather than a chronic disease. (I also would double check that these diagnoses are definitely from before 2020 and not during the pandemic)

R2.2 The reviewer is right, that the term 'symptoms of malaise and fatigue' is unspecific, but we observed a persistent and strong associations after accounting for surrogates of healthcare utilization and multimorbidity along with other risk factors (adjusted HR: 1.66, 95%-CI: 1.58 - 1.74, p-value=7.3x10⁻⁹²). We further like to highlight that the association between symptoms of malaise and fatigue and a higher risk for, at least, long COVID have also been reported by Zang et al. Nature Communications 2023 in a large US-based medical record database. These

findings might still be explained by reverse causation, that is, patients reporting malaise and fatigue already before the pandemic are now diagnosed with long COVID due to higher awareness of clinicians for related diseases. However, our finding that the same symptom set is independently and genetically linked to severe COVID-19 provides support for the hypothesis that there seems to be some kind of general susceptibility to viral induced long-term health effects among patients.

Notably, diseases most strongly correlated with symptoms of malaise and fatigue in our disease-disease network, such as anxiety disorders or hypothyroidism, were less strongly associated with severe COVID-19.

However, we agree with the reviewer that the evidence we provide should be treated as exploratory, which we now clearly point out in the revised discussion section of the manuscript (p10, lines 328-330).

3. As a reader I'm somewhat confused about the divergent effect of certain genetic loci and the authors explanation for this is that those carrying a high-risk allele may adhere to more cautious measures? If that was the case, shouldn't the diseases also show protective effects? For example, shouldn't we see that the ICD codes for pulmonary fibrosis are also associated with protection? Can the authors further justify or explain this finding.

R2.4 *We were, and still are, somewhat puzzled by this finding, too. We agree with the reviewer that the previous explanation by others (like Fadista et al. EBioMedicine 2021) contradicts our and other observational studies by which patients with pulmonary fibrosis are at a substantially increased risk for severe COVID-19.*

In general, this finding might well be the result of multiple, possibly independent factors. Firstly, opposing effects of a shared genetic risk variant have been described for autoimmune but also other diseases (e.g., Ellinghaus et al. Nat Gen 2016), with the general interpretation that genetically determined fine-tuning of the immune response can predispose to different diseases, while the onset of one, or many others, might 'protect' the onset of another. However, no sound mechanism has been proposed so far.

Secondly, the hazard ratios we and others report should be considered as conditional probabilities, being the product of the true underlying disease-increasing risk and the probability and duration of viral exposure. Since patients with severe respiratory disease have been shielded early in the pandemic, they may well be underrepresented even among patients with severe COVID-19. Such an effect can introduce ascertainment bias in genetic association studies, by which genetic variants strongly associated with selection, like the MUC5B locus for pulmonary fibrosis, appear as protective signal in the outcome (e.g., Solovieff et al. Nat Rev Gen 2014). However, since genetic risk loci act independently of each other, such ascertainment bias can be restricted to only a subset of loci.

Lastly, the same genetic variant may act in a cell-type, and more generally context-dependent, manner on different diseases. For example, while we observed multiple loci shared between asthma and COVID-19, including those in opposing directions, we did not observe a significant genetic correlation and one might hypothesize that hyperactive immune pathways that predispose to asthma at an early age protect from severe viral infections later in life. We added these different explanations to the revised version of the manuscript (p11, lines 338-347).

4. Some of your Results include topics that should be in the Discussion. For example, I would move Lines 246-250 and 263-267 to the discussion

R2.5 We followed this helpful recommendation of the reviewer and shifted sections from the results to the discussion (p11, lines 338-340). However, we hope that the referee agrees with us not shifting parts of the TYK2 presentation to the discussion, as otherwise important contextualization why this discrepant finding is of high interest would be missing.

Minor Point

1. Its interesting that the respiratory failure endpoint has less total patients than deaths due to COVID-19. While this is likely in part due to the need for a positive test and the diagnosis code for the resp failure, I also wonder if certain diagnosis codes were excluded that should have been included (for example, J96.01 – Acute respiratory failure with hypoxia, J96.02...). However, I am listing this as a minor comment, because I doubt their exclusion really effects the findings of the author’s study, but I wanted to point it out for their future epidemiology studies.

R2.6 We thank the reviewer for pointing additional codes out, but as the reviewer already suspected, adding those additional codes did not change the number of patients with severe respiratory failure substantially (N=680 compared to 662).

2. In Results: Line 111, fix the date

R2.7 We corrected the typo accordingly.

3. Supplementary Table 2? Change to landscape and reformat. It it’s current format it is very challenging to read.

R2.8 We suppose that the poor reading of ST2 must have been due to a conversion problem during the submission process and now provide ST2 as a clean Excel file. We like to note, that having the information for one disease across all COVID-19 related outcomes within the same row is key to understanding our findings.

4. Seems like supplementary table only has supplement table 2, although I see others in the excel file?

R2.9 We apologize for this formatting error and now provide one Excel file with all relevant tables.

5. Line 114 - What are the two exceptions? Also can you clarify what you mean here? Is this of all ~1,400 diseases evaluated?

R2.10 We clarified this section, stating that only two diseases were associated with a decreased risk for severe COVID-19 (p4, line 159-162).

6. Line 400 , two a’s

R2.11 Thank you! We have now corrected this typo.

7. Line 235, 49 known genomic regions. Can you cite where this is coming from?

R2.12 Since we used genome-wide summary statistics excluding UK Biobank from the COVID-19 HGI, we performed regional clumping to select independent regional lead signals as input for colocalization analysis. We added this information to the revised version of the manuscript (p17, lines 536-538).

8. Line 408 can the author include definitions for Y2b89, Y2b8a, Y2b87, and Y2b88.

R2.13 We added the corresponding explanations (p13, line 404-406).

Reviewer #1 (Remarks to the Author):

The authors have addressed my comments appropriately, and notably improved their manuscript.

I do not have any further suggestions.

Reviewer #2 (Remarks to the Author):

The authors have sufficiently addressed all of my comments. They have developed a well-conceived and thoughtful manuscript.

REVIEWERS' COMMENTS

Reviewer #1 (Remarks to the Author):

The authors have addressed my comments appropriately, and notably improved their manuscript.

I do not have any further suggestions.

We thank the reviewer for the careful review of our paper and the helpful suggestions.

Reviewer #2 (Remarks to the Author):

The authors have sufficiently addressed all of my comments. They have developed a well conceived and thoughtful manuscript.

We thank the reviewer for the careful review of our paper and the helpful suggestions.